# Galactomannan Produced by *Aspergillus fumigatus*: An Update on the Structure, Biosynthesis and Biological Functions of an Emblematic Fungal Biomarker

**DOI:** 10.3390/jof6040283

**Published:** 2020-11-12

**Authors:** Thierry Fontaine, Jean-Paul Latgé

**Affiliations:** 1Unité de Biologie et Pathogénicité Fongiques, Institut Pasteur, 25 rue du Docteur Roux, 75015 Paris, France; 2Institute of Molecular Biology and Biotechnology FORTH and School of Medicine, University of Crete Heraklion, 71003 Crete, Greece; jean-paul.latge@pasteur.fr

**Keywords:** galactomannan, *Aspergillus fumigatus*, cell wall, galactofuranose, glycosyltransferase, polarized growth, immune response

## Abstract

The galactomannan (GM) that is produced by the human fungal pathogen *Aspergillus fumigatus* is an emblematic biomarker in medical mycology. The GM is composed of two monosaccharides: mannose and galactofuranose. The furanic configuration of galactose residues, absent in mammals, is responsible for the antigenicity of the GM and has favoured the development of ELISA tests to diagnose aspergillosis in immunocompromised patients. The GM that is produced by *A. fumigatus* is a unique fungal polysaccharide containing a tetramannoside repeat unit and having three different forms: (i) membrane bound through a glycosylphosphatidylinositol (GPI)-anchor, (ii) covalently linked to β-1,3-glucans in the cell wall, or (iii) released in the culture medium as a free polymer. Recent studies have revealed the crucial role of the GM during vegetative and polarized fungal growth. This review highlights these recent data on its biosynthetic pathway and its biological functions during the saprophytic and pathogenic life of this opportunistic human fungal pathogen.

## 1. Introduction

The galactomannan (GM) that is produced by the human fungal pathogen *Aspergillus fumigatus* is an emblematic biomarker in medical mycology. *A. fumigatus* is the Aspergillus species most frequently involved in human disease in developed countries. Global multinational prevalence of aspergillosis reaches an estimated 3,000,000 cases per year of chronic pulmonary aspergillosis and 250,000 cases per year of invasive aspergillosis [1]. In the 1970s, the galactomannan polymer was identified as a circulating molecule in the biological fluids of immunocompromised patients that were infected with Aspergillus [2,3]. The identification of rat monoclonal antibodies against the GM has led to the development of a sensitive diagnositic enzyme immunoassay [4]. The GM is composed of two monosaccharides: mannose and galactofuranose. The furanic configuration of galactose residues, absent in mammmals, is responsible for the antigenicity of the GM. Despite the presence of false positive and negative detection and despite other available biomarkers, such as β-1,3-glucans or the development of a PCR method, the GM remains the unavoidable biomarker for the IA diagnosis [5,6] and has led to the recent development of new monoclonal antibodies and the lateral flow assay (LFA)-GM technology [7,8].

Although the GM polymer was described more than 50 years ago, the biological functions and biochemical pathway that are responsible for its biosynthesis remain insufficiently understood. This review highlights the last data on the GM produced by *A. fumigatus* and shows that the GM play an essential role for fungal apical growth and host-pathogen interactions.

## 2. GM: A Unique Fungal Specific Polysaccharide

In 1994, Latgé and his colleagues have described the chemical structure of a soluble 20 kDa GM produced by *A. fumigatus*, which was isolated from the culture supernatant. Carbohydrate analyses of this extracellular polymer showed that the GM was composed of a linear mannan core with an α-1,2-linked mannotetraose repeating unit attached via α-1,6-linkage. Side chains that are composed of an average of 4 to 5 β-1,5-galactofuranose units are linked to C-6 and C-3 positions of α-1,2-linked mannose units of the mannan (Figure 1). The complete chemical structure of the GM, which was initially elucidated by Latgé et al. [9], was completed over the years. NMR development for biomolecules and chemical carbohydrate synthesis allowed for updating the GM structure. The mannan chain structure containing a repeat unit has been confirmed, but galactofuran side chains appeared more variable and complex. The presence of β-1,6-glycosidic intrachain linkage has been now confirmed [10,11]. The size of galactofuranose side chain is highly dependent on the growth conditions and it could be up to 10 residues [9,10]. The apparent molecular weight of GM varies from 15 up to 50 kDa according to the variation of the galactofuranose content [10].

The presence of such mannan structure was described in secreted fungal polymers isolated from other *Aspergillus* species and other ascomycetous species such as *Penicillium* or *Trichophyton* [12,13,14,15]. This specific mannan sequence is totally unrelated to the mannan polymers that were found in yeast. *Saccharomyces cerevisiae* mannans result from the elongation of N-glycan and are composed of an α-1,6-mannan linear chain with side chains of α-1,2-mannoside of one to five residues with or without a α-1,3-mannoside end capping [16]. The use of strong chemical degradation, i.e., hydrazinolysis and nitrous deamination to purify the *A. fumigatus* GM from the culture medium did not allow for demonstrating its linkage to putative protein carrier. Other studies from Latgé’s group have shown that (1) GM is part of the fungal cell wall where it is covalently linked to β-1,3-glucan [17] and (2) GM is bound to cellular membrane through a glycosylphosphatidylinositol (GPI)-anchor (called lipo-GM)[18]. The specific degradation by acetolysis of these three species of GM bound to the plasma membrane or the cell wall or secreted has confirmed the presence of the same tetramannoside repeat unit in all these molecules. These two later forms of GM were described in *A. fumigatus* in both conidia and mycelium. The GPI-lipid anchor is an inositol-phosphoceramide (IPC), where the ceramide is composed of a C_18_-phytosphingosine with an N-linked 2-hydroxyfatty acid. In fungi, the GPI glycan moiety is mainly composed of a tetra-α-mannoside, being identical to the repeat unit of the GM, which is linked to a glucosamine residue. The glycan moiety may be capped by an α-1,3-linked mannose residue [19,20]. The GM chain is linked to the IPC anchor via mannan chain to the glycan core of GPI.

GM from the culture supernatant has been suggested to be the N-glycan moiety of glycoproteins in *A. fumigatus* [10]. However, in this later investigation, the demonstration of covalent linkage between protein and GM through PNGase or protease digestions were missing. Other studies have described glycoprotein extracellular antigens with molecular weights of between 20 to >200 kDa, which were positively labeled by an anti-GM MAb [21,22,23,24]. Glycan moities containing galactofuranose residues could be N- or O-linked to proteins [10,22,23,24,25], complicating the studies in the identification of the nature of the real circulating antigen. In addition, different investigations of N-glycans from purified glycoproteins [23] or crude glycoprotein fractions have shown that *A. fumigatus* mycelium produces relative short N-glycan containing five up to 11 hexose residues [26,27]. No larger N-mannan with a 20 kDa size has been identified to be linked to protein in these studies, showing that the presence of mannotetraose repeating unit in glycan structure linked to a carrier protein remains to be demonstrated. In addition, because the presence of a membrane-bound GPI anchored GM was identified, the presence of a lipid anchor on the GM released the culture surpernatant has been also investigated by chromatographic purification on octyl-sepharose. In contrast to the lipo-GM isolated from membrane preparation, the GM that is released in the culture medium does not possess a lipid-anchor [28].

Taking together all the data on the GM structure, GM that is produced by *A. fumigatus* is a unique fungal polysaccharide having three different forms: membrane bound through a GPI-anchor, cell wall localized GM where it is covalently linked to β-1,3-glucans and released in the culture medium as a free polymer (Figure 1).

## 3. Biosynthesis of the GM in *A. fumigatus*

The structural cell wall polysaccharides (chitin and β-1,3-glucan) are synthesized at the plasma membrane level by protein complexes [29]. In contrast, the GM is intracellularly synthesized, where Golgi transporters of GDP-Man and UDP-Galf are required for the GM polymerisation.

### 3.1. What Are the Initial Steps Leading to Lipo-GM Biosynthesis?

The similarity of the carbohydrate and lipid moieties of the lipo-GM with thoses of GPI-anchored proteins (GPI-APs) suggested that the GM might follow the GPI biosynthetic pathway. The GPI synthesis pathway occurs at the endoplasmic reticulum (ER) membrane. Nine biochemical steps are required for the complete synthesis of the intermediate prior to being transferred to the targeted protein: transfer of N-acetylglucosamine (GlcNAc) onto a phosphatidylinositol (PI), GlcNAc deacetylation, inositol acylation, and the addition of four mannose and three phosphoethanolamine residues. This pathway is conserved in filamentous fungi, including in *A. fumigatus* [30,31]. After the transfer to the protein bearing the C-terminal signal sequence for GPI attachment and before the exit of ER, a GPI remodeling occurs to modify the lipid moiety and remove the first two phosphoethanolamine groups. In fungi, a specific lipid remodeling leads to the substitution of the diacylglycerol by a ceramide [32] and occurs in three steps where the de-O-acetylation by the phospholipase A2 activity of Per1p is a key step [33]. A comparative structure analysis by the mass spectrometry of lipid anchor of GPI-anchored proteins (GPI-APs) and lipo-GM was performed with the *Δper1* mutant in *A. fumigatus* [34]. The *∆per1* mutant was still producing the Lipo-GM and its lipid anchor contains an inositol-phospho-ceramide (IPC) identical to the one of the parental strain; whereas, the GPI-APs of *Δper1* mutant only contained inositol-phosphodiacylglycerol as lipid anchor. The defect on GPI lipid remodelling has no effect on the lipid anchoring of GM, showing that the biosynthesis of the Lipo-GM is independent of the biosynthetic pathway of the GPI-anchor. 

Another possibility was that the synthesis of the Lipo-GM could follow a similar pathway to the synthesis of glycosyl-inositolphosphoceramides (GIPC) where monosaccharides are sequentially added to the IPC anchor. Two types of GIPC are produced in *A. fumigatus*, acidic and zwitterionic GIPC [35,36] and their synthesis, are independent of *PER1*. The biosynthesis of GIPCs starts with the addition of glucosamine or mannose residue onto the inositol ring of the IPC [37,38]. However, the inactivation of mannosyltransferase (MitA) and N-acetylglucosaminyltransferase (GntA) required for the GIPC synthesis has no effect on the Lipo-GM synthesis [37,38]. This result indicated that the Lipo-GM synthesis is also independent of GIPC synthesis and it required specific glycosyltransferase activities unidentified to date.

### 3.2. Synthesis of the Mannan Chain

The polymerisation of the mannan chain of the GM requires α-mannosyltransferases, and most of them use the GDP-Man as sugar donor. The first study describing a defect in GM synthesis in *A. fumigatus* resulted from the deletion of the GDP-Man transporter, Gmt1 [39]. *GMT1* is the heterologous gene of *VRG4* in yeasts [40,41]. Vrg4 allows for the transport of GDP-Man from the cytosol into the lumen of the Golgi where numerous glycosyltransferases are located. It is essential for the full N- and O-mannolysation of glycoproteins and the GIPC synthesis. In *A. fumigatus*, the deletion of *GMT1* leads to the absence of Lipo-GM in cellular membrane and cell wall-GM, showing that the biosynthesis of the GM starts in the Golgi apparatus [39].

In silico analysis of the genome of *A. fumigatus* has allowed for the identification of 23 putatives Golgi α-mannosyltransferases. Most of them have orthologs in yeasts. Among investigations of the function of mannosyltrasferases in *A. fumigatus*, an undecuple multiple mutant has been constructed, where α-1,2 or α-1,6-mannosyltransferase members of CAzy GT32 (Och orthologs), GT34 (α-1,6-mannosyltransferases), GT62 (Mnn9, Anp1, and Van1 orthologs), and GT71 (α-1,2-mannosyltransferases) were deleted [42]. No difference in growth and GM content have been observed in this multiple mutant, showing that no-one of these gene are essential for GM synthesis. Recently, two independent research teams have shown that two α-mannosyltransferase members (CmsA/Ktr4 and CmsB/Ktr7) of the GT15 family (Mnt1/Kre2 family) are essential to the GM synthesis in *A. fumigatus* [43,44]. Both single and the double mutants produce a low amount of GM released in the culture medium. NMR data showed that core-mannan signals were absent or substantially truncated, indicating that core-mannan structures are altered and/or lost in the absence of CmsA/Ktr4 or CmsB/Ktr7 [43]. At the cell wall level, the deletion of both genes led to the absence of GM cross-linked to β-1,3-glucan [44].

In vitro enzymatic analysis of recombinant CmsA/Ktr4p has shown that it carries an α-1,2-mannosyltransferase activity, which suggested that the two other members of GT15 family in *A. fumigatus* possess the same activity [43,44]. Ktr1p is not required for the GM synthesis and did not compensate the deletion of the two others, showing differences between homologous enzymes in the same CAZyme family [44,45]. The *CMSA*/*KTR4* and the *CMSB*/*KTR7* genes are both essential to the GM synthesis, but are not redundant. Our current hypothesis is that CmsA/Ktr4p and CmsB/Ktr7p are α-1,2-mannosyltransferases acting sequentially during the synthesis of the tetra-α-1,2-mannoside unit. The product of the first transferase activity would then be the acceptor of the second one and the recognition of the acceptor by the transferase activity represents a critical point for driving the sequential addition of α-1,2-mannoside residue [44]. This hypothesis also suggests the requirement of a transglycosidase/transferase able to polymerase tetramannoside oligomers through an α-1,6 linkage. Such a type of enzyme activity remains to be identified.

Surprisingly, a lipo-GM fraction was still produced by *ΔcmsA/ktr4* and *ΔcmsB/ktr7* mutants. However, the analysis of the lipo-GM structure clearly showed that this lipo-GM possesses a totally altered mannan structure that was characterised by the absence of the classical tetramannoside repeat unit found in the native GM (Figure 2A). GIPC analysis showed that the deletion led also to an increase in the size of the carbohydrate moiety of the GIPC (Figure 2B). These data clearly showed that the deletion of *KTR4* and *KTR7* led to the absence of synthesis of the cell wall GM and induce the synthesis of new mannosylated structures, probably due to the compensation by other unknown mannosyltransferase activities.

The number of mannosyltransferase are functional in *A. fumigatus* [37,42,45,46,47]. In *S. cerevisiae*, Ktr orthologs belong to the GT15 family in the CAZy database and have been characterised as α-1,2-mannosyltransferases playing a role in N- and O-mannosylation of glycoproteins [48,49,50,51]. Although *A. fumigatus* AfKtr4p and AfKtr7p share high sequence homologies with the yeast Ktrp orthologs and display α-1,2-mannosyltransferase activity, the deletion of *A. fumigatus KTR4* and *KTR7* genes did not alter protein mannosylation, but led to the loss of the galactomannan polysaccharide. These data provide another clear example of the finding that orthologous genes code for proteins that may have very different biological functions in yeasts and filamentous fungi, even though they have very related enzymatic activities, at least in vitro.

### 3.3. Addition of Galactofuran Side Chains

Galactofuran side chains are dependent of several activities that are involved in the synthesis of UDP-galactofuranose (UDP-Galf), the transport of the nucleotide sugar and then the elongation of the galactofuran chains. UDP-galactopyranose mutase is a key enzyme in UDP-Galf biosynthesis in microorganisms. Such enzyme activity that was able to convert the pyranic configuration into furanic one was described for long time in procaryotes and described in eucaryotes (*Aspergillus* and *Leishmania)* for the first time in 2005 [52]. Only one *UGM1/GLFA* gene is present in *A. fumigatus* genome and its deletion led to the total absence of galactofuranose containing molecules, such as galactose moieties of glycoproteins, GIPCs, and the GM polymers [53,54]. Interestingly, the absence of UDP-Galf did not prevent the synthesis of the mannan chain of the GM, neither its integration into the cell wall showing that the polymerisation of the repeat mannan unit is not dependent on the presence of galactofuranose side chain.

The first identification of a nucleotide-sugar transporter specific for UDP-Galf has been identified in *A. fumigatus* [26]. The protein was called GlfB because of its implication in galactofuranose metabolism and it was selected from its homology to other members of the nucleotide-sugar transporter (NST) family, its location on chromosome 3 directly downstream of the *UGM1/GLFA* gene. It is a 400-amino acid protein with 11 predicted transmembrane helices. It is a very distinct member of the NST family, because NSTs classically exhibit 8–10 predicted hydrophobic domains with both the N and C terminus situated on the cytoplasmic side of the Golgi. The homolgous gene of *AfUGM1*/*GLFA* are present in other fungal species [55,56]. Deletion of the *GLFB* in *A. fumigatus* or homologs in other *Aspergillus* species leads to a similar phenotype as the *UGM1/GLFA* deletion. No more galactofuranose residues are found in any carbohydrate structures [26], showing that only one NTP-sugar is sufficient to specifically transport UDP-Galf into the Golgi where galactofuranosyltransferases are localized.

The first fungal β-galactofuranosyltransferase GfsA that has been identified in *A. nidulans* and *A. fumigatus* is a Golgi type II transmembrane protein [57]. This member of the CAzy GT31 family is involved in the addition of galactofuranose residues to mannan structure. In vitro, it was shown that the recombinant AfGfsA acts as a UDP-α-d-galactofuranose: β-d-galactofuranoside β-1,5-galactofuranosyltransferase involved in the biosynthesis of galactomannans [58]. *A. niger*, as well as *A. nidulans* and *A. fumigatus*, contain three β-galactofuranosyltransferases encoding genes in their genomes [58,59]. The deletion of *AfGFSA1* leads to the absence of detection of secreted glycoproteins by an anti-Galf MAb and a reduction of 84% of β-1,5-galactofuranoside residues in the GM [58], showing that the UDP-Gal mutase, transporter specific for UDP-Galactofuranose, and β-galactofuranosyltransferase are equally essential for obtaining a correct β-galactofuranosylation.

### 3.4. Cross-linking to Cell Wall β-1,3-Glucan

The cell wall GM is cross-linked to β-1,3-glucans and its amount is dependent on the growth conditions and morphotypes analysed (conidia versus mycelium) [60,61]. Recently, we have shown that DFG members of the CAZy GH76 family are essential for the covalent anchoring of the GM to the cell wall [28]. AfDfgp are homologs of yeast Dfg5 and Dcw1 proteins. The increased release of GPI-anchored proteins in the *dfg5* mutant or the *dcw1* mutant in yeast suggested that these proteins were involved in the cross-linking of GPI-CWPs to the cell wall glucans [62,63]. In *A. fumigatus*, seven members of Dfg proteins were identified. Single gene deletion suggested that Dfg3 was the most important member of the Dfg family. However, all members of the family were involved in the cross-linking process. Since the total loss of GM bound to β-1,3-glucans requires the successive sequential deletion of all members of the family. However, the *Δdfg* mutants still produced membrane-bound GM with the same chemical structure as the one of the parental strain, which showed that Dfg proteins are not involved in GM biosynthesis, but that these Dfg proteins only play a key role in the cross-linking of GM to β-1,3-glucans [28].

Two bacterial members of GH76 family have an α-1,6mannosidase activity [64,65]. In fungi, no mannan hydrolytic activity was identified while using recombinant Dfg protein. However, the Yeast *DFG5* deletion mutant could be complemented by the *AfDFG3* showing that yeast and filamentous homologs have similar function, even though the structures transferred would be different since there is no GM in yeast and the yeast cell wall mannan are not cross-linked to β-glucans but are the N-glycan moiety of cell wall glycoproteins [16]. Since the lipid-anchor of fungal GPI-APs and Lipo-GM are similar, it was also tempting to speculate that such Dfg proteins were involved in the cross-linking of membrane GPI-anchored molecules onto cell wall glucan. However, in both yeast and filamentous *Δdfg* mutants, GPI-anchored and non-GPI cell wall mannoproteins were both released, as has been frequently seen in many other cell wall mutants [28,62,66,67]. Because all attempts to show a transglycosidase activity in vitro with recombinant proteins have failed to date, the biochemical function of Dfg orthologs remains unknown.

### 3.5. Conclusion on the GM Biosynthetic Pathways

Figure 3 summarizes the biosynthetic pathway of the GM in *A. fumigatus*, which remains unique in the fungal kingdom. This pathway required (1) the synthesis of nucleotide-sugar that occurs in the cytosol and their specific transport into the lumen of the Golgi, (2) the polymerisation of the mannan chain on a lipid acceptor, probably an IPC, and the addition of galactofuranose side chain, (3) the trafficking of the lipo-GM to the external face of the plasma membrane, and (4) the translocation onto the cell wall β-1,3-glucans where Dfgp play a key role.

All of the glycosyltransferases that are involved in the lipo-GM synthesis are not yet identified. The N-acetylglucosaminyltransferase and the consecutive *N*-deacetylase required for the formation of GlcN-IPC lipid-anchor are unknown. Based on its structure with a repeat unit, our current hypothesis that the biosynthesis of the mannan chain should be processed in two steps: the synthesis of the tetra-α-1,2-mannoside repeat unit, where CmsA/Ktr4p and CmsB/Ktr7p are essential, and the polymerisation of the repeat unit. Such a process described in procaryotes required specific enzymes and acceptor [68,69] that are not yet identified in *A. fumigatus*. The release into the culture medium is independent of Dfg proteins [28], but other enzyme activity, such as phospholipase or glycohydrolase, is required to remove the lipid anchor.

## 4. Functions of the GM

The functions of the GM during the fungal life and infections remain insufficiently understood. Recent discoveries on the mannan biosynthesis and the construction of GM deficient mutant have highlighted the importance of this unique fungal polysaccharide.

### 4.1. A Cell Wall Without Galf Has an Altered Structure

Phenotype analysis of the galactofuranose deficient *Aspergillus* mutants brought some clues of the biological functions of galactofuranose containing molecules in the fungal life. In *Aspergillus* species, phenotypes were first described in UDP-Gal mutase deficient strains. In *A. niger* and *A. nidulans*, the absence of galactofuranose induces a slight growth reduction with an increase sensitivity to temperature as well as cell wall disturbing compounds, such as calcofluor white. Respective mycelium are hyperbranched, which suggests a role of galactofuranose residues in the cell wall organisation and permeability [55,70]. In *A. fumigatus*, similar phenotypes were observed but were less pronounced [53,54]. However, the weak phenotype seen for these mutants may result from compensatory reactions due to other cell wall associated enzymes. The deletion of *UGM1* led to a cell surface alteration inducing an increase of adherence to abiotic and biotic surface and an increase of hydrophobicity. Particularly, the presence of galactofuranose was initially thought to be responsible for the reduction of the adherence properties of mannan chain to plastic and glass solid surface [53]. This change in adhesion properties was in fact due to the unmasking of the cell wall mannan, but also to a complete modification of the surface structure resulting from the overproduction of galactosaminogalactan [53,71]. In *A. niger*, the absence of galactofuranose synthesis is compensated by an overexpression of α-1,3-glucan synthesis [56,72], which makes it difficult to conclude on exact functions of galactofuranose residues in *Aspergillus* species. The phenotype of mutant deficient in UDP-Galf transporter or in β-galactofuranosyltransferase share similar phenotypes (reduced growth, aberrant branching morphology, reduced conidiation, increased sensitivity to CFW, and increased expression of *AGSA*) [59].

### 4.2. Mannan Controls Polarized Growth in Aspergillus

The absence of GDP-Man transporter (Gmt1) led to a strong defect in growth of AfS35 wild type *A. fumigatus* [39]. While using the heterokaryon rescue methodology, we showed that *GMT1* is an essential gene in the *A. fumigatus* akuB CEA17 genetic background (Lambou and Latgé, personnal communication), showing that the importance of mannosylation in *Aspergillus* depends of the fungal strain genetic background.

The phenotype of the α-mannosyltransferase mutants depends strongly of the gene deleted and suggested that these enzymes have a different function probably linked to their respective substrate acceptors. The deletion of Golgi mannosyltransferases involved in N-mannan elongation of glycoproteins or in GIPC synthesis have no or weak effect on filamentous growth [37,42,45,46,47]. In *A. fumigatus*, N-mannan elongation play a major role at the conidia level, where multiple deletion of α-mannosyltransferase genes led to conidial cell wall alteration and permeability, showing the presence of different mannan polymers, depending on the morphotype [42]. In agreement with the essentiality of GDP-Man transport into the Golgi, the deletion of the *CMSA/KTR4* or *CMSB/KTR7* led to a severe phenotype. *Δktr4* and *Δktr7* mutants both have strong defects in vegetative growth, conidiophore formation and conidiation [43,44]. Mycelium of these mutants were hyperbranched with formation of swollen filaments (balloon morphology) after 48 h of growth showing altered cell wall organisation. At the conidia level, the absence of GM synthesis led to enlarged and fragile conidia and to a mis-regulation of the germination process where the polarization of germ tube formation was altered [44].

GM is part of the structural cell wall skeleton, where it is covalently linked to the chitin-β-1,3-glucan complex. A defect in the cross-linking of GM into the cell wall of *A. fumigatus* that resulted in the deletion of *DFG* family members led to a strong defect in conidiation and hyperbranched reduced mycelial growth [28]. Although the GM is still present at the plasma membrane level in DFG mutants as in WT, its absence of covalent linkage in the cell wall was not compensated by the higher production of chitin (a compensatory reaction often seen in cell wall mutants), showing that the formation of the linkage of GM to the core chitin-β-1,3-glucan plays an essential role in the vegetative growth. In contrast, the absence of synthesis of other cell wall polymers (α-glucan, β-1,3/1,4-glucan, galactosaminogalactan (GAG)) did not alter the filamentous polarized growth [71,73,74].

### 4.3. GM Induces a Host Immune Response

The fact that GM is secreted in the infected patient suggested that this polysaccharide may induce a specific response of the host [75,76]. Indeed, in a vaccination mouse model of aspergillosis, the intranasal administration of purified GM or Lipo-GM failed to induce protection, which was concomitant to a (IL-4/Gata3), and Th17 (IL-17A and F/Rorc) activation, which was independent to the presence of the lipid anchor of the GM [77]. However, very few studies analysed the role of GM during the pathobiology of aspergillosis in the immunocompromised and immunocompetent host [29,78]. The recognition of GM by PRR remains particularly poorly understood. 

The first step in the induction of the immune response is the recognition of the GM by host cells. How is GM recognised by host cells? Toll-like receptors (TLRs) and C type lectins are major players in the recognition of mannosylated structures by the host [79,80]. The role of TLR 2 and 4 in the recognition of *A. fumigatus* cells has been shown [81,82,83,84]. GM immunomodulates the proinflammatory responses, mainly through the TLR4 signalling [85]. However, the mechanisms by which TLR4 and TLR2 recognise mannose-containing glycoconjugates and other fungal polysaccharides are still unknown [86]. The finding that Th2-type cytokines increase the susceptibility to invasive aspergillosis has prompted studies on the role of dentritic cells (DC) during *Aspergillus* infections [87,88]. DCs and macrophages bind and internalize *A. fumigatus* conidia via the DC-SIGN receptor, and this binding can be inhibited by GM isolated from *A. fumigatus* [89,90]. The use of sugar-conjugated nanoparticle showed that the β-galactofuranose residues are recognised by DC-SIGN [91]. In vitro, fungal-like particles decorated with galactomannan isolated from *A. fumigatus* are recognised by macrophage and induced TNF-α production through Dectin-2 [92]. The presence of α-1,2-mannoside sequence in the GM structure is in agreement with its interaction with the Dectin-2 [80]. *In vivo*, the investigation of Dectin-2 and MMR expression both at transcriptional and translational levels in human lung revealed that Dectin-2 and MMR proteins which are expressed relatively low level in normal lung samples exhibited marked overexpression in *A. fumigatus*–invaded lung tissues in both ABPA and invasive aspergillosis patients [93]. It is still unknown whether GM plays a major role in this immune response. The interaction between human plasmacystoid dentritic cells (pDCs) and *A. fumigatus* hyphae is mediated through Dectin-2 [94]. This interaction that triggers antifungal activity and cytokine release was partially inhibited by anti-Dectin-2 antibodies and yeast α-mannans [94]. Although it has been shown that MMR interacts with *A. fumigatus* [95], the biochemical recognition of GM by MMR remains to be investigated. 

Other PPRs may interact with fungal mannans. Mannose binding lectin (MBL) and other collectins (SP-D and SP-A) play plausible role in pulmonary defence against *A. fumigatus* [21,96,97,98]. The pulmonary surfactant protein D (SP-D) recognises GM in a calcium dependent manner, whereas MBL and SP-A show, respectively, a weak or the absence of binding to GM [99]. In addition, *A. fumigatus* produces a number of glycoproteins and glycosphingolipids containing α-mannoside residues and an unique cell wall mannan in *Aspergillus* conidia [42]. All of these carbohydrate structures may interact with the same PRRs as GM, making difficult to understand the true in vivo function of GM during infection. 

The tools available to analyse the host immune response only start to be developed. In the past, some authors used erroneously commercially available GMs, which are isolated from plants. These plant GMs are structurally unrelated to the one that was produced by *A. fumigatus* and make conclusions associated to the role of GM totally wrong and inappropriate. Using GM mutants to assess the role of GM during the infection is also difficult, because the cell wall of these mutants may be modified to compensate for the absence of GM and these modifications may be more important for the fungal virulence than the lack of the GM itself. As example, GM deficient mutants, which are less virulent, have a slow growth and produce a high amount of cell wall chitin [44] and galactofuranose deficient mutant overproduces GAG, which is a virulence factor [71]. In addition, the virulence of *ΔUgm1/GlfA* mutant in a mouse infection model of invasive aspergillosis seems to also depend on the genetic background of the *A. fumigatus* parental strain [53,54]. Chemically synthesized oligosaccharides are essential tools that allow for precisely deciphering the specificity of the immune response. Using this system, it was shown that the antibodies recognised the terminal disaccharide Galf-β-1,5-Galf [100]. The same Galf-β1,5-Galf dissacharide is the minimal size that is required to induce the production of the specific cytokines, interleukin 1 beta (IL-1), IL-1Ra, IL-6, produced by peripheral blood mononuclear cells (PBMCs). In contrast, the mannan moiety of the GM does not induce antibody or cytokine response [100].

## 5. Perspectives

The GM that is produced by *A. fumigatus* is a unique fungal polysaccharide. Although it has been described 50 years ago, recent discoveries on its biosynthesis highlight its importance for the vegetative growth and for the fungal cell wall organisation. The overall functions of this polysaccharide are summarized in Figure 4. Based on this prevalence of aspergillosis infections, future investigations will be operated under three axes:(1)Improvement of diagnostic test. Earliers detection of aspergillosis is required to improve antifungal treatment and eliminate fungal burden. The GM diagnostic test is based on the detection by ELISA of galactofuranose side chain. According to the carbohydrate structure of the GM, a double detection that is based on the development of antibodies on the specific mannan chain and side chain of galactofuranose may improve the specificity of detection. Galactofuran chains are part of the GM as well as of glycoproteins and GIPC. Carbohydrate structures that are produced by *A. fumigatus* during infection have never been investigated. Such studies of the GM and other cell wall polymers released during infection may identify new aspergillus biomarkers.(2)GM biosynthesis may be a new drug target. The strong defect of vegetative growth in mannan deficient mutant in *A. fumigatus* shows the essential role of the GM inside the cell wall, particularly in the organisation of the structural core. The cross-linking of the GM to the β-glucan-chitin complex is also an essential event for the polarized fungal growth. The biosynthetic pathway of the GM includes two crucial steps for the vegetative growth, GDP-mannose transport into the lumen of the Golgi, and the mannan polymerisation. Because Gmt1p transporter and α-1,2-mannosyltransferase members of the GT15 family are absent in human, the development of specific inhibitors of both activities that are essential to GM polymerisation are new approaches to design new anti-*Aspergillus* drugs. The main question is to understand how this GM, which is present at relatively low concentration in the cell wall, is so crucial. The GM is originally a soluble polymer and its organisation and interaction with the other polysaccharides of the structural core are unknown. The cell wall GM is present under two forms: bound to the outer layer of the plasma membrane and covalently linked inside the cell wall. One of our hypotheses is that the GM may be required for the interaction between the plasma membrane and cell wall, essential to polarized fungal growth(3)What is the role of a circulating polysaccharide in the systemic immune response of the host? It is essential to define the immune receptors interacting with Aspergillus GM and driving the host reponses during infections. Although the fungal mannans usually play a critical role in innate host response, the mannan of the GM is not recognised by the human antibody. This lack of recognition has not been investigated to date. Deep investigations using GM mutants as well as purified GM in combination or not with other cell wall components are required to better understand the role of GM during *Aspergillus* infection and host immune response.

## Figures and Tables

**Figure 1 jof-06-00283-f001:**
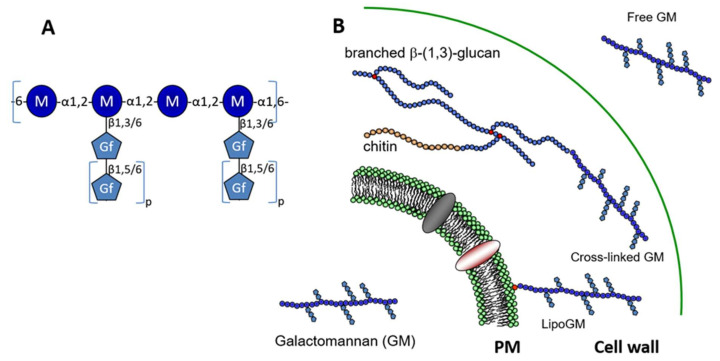
Structure and localisation of the galactomannan. (**A**) Structure of the repeat unit of the galactomannan (GM). The main mannan chain is composed of a tetra-α-1,2-mannoside connected through a α-1,6 linkage. Side chains are composed of galactofuranose residues linked to mannose through a β-1,3 or β-1,6 linkage. Depending on the growth condition, variations of galactofuranose content and glycosidic linkage were observed (0.2 to 2.6 galactose residues per mannose residue were observed [9,10]. The size of side-chain is variable from 1 up to 10 galactofuranose residues (0 < *p* < 10), where galactofuranoses were linked in β-1,5. The presence of β-1,6 were also described in some growth condition where it represents up to 10% of total β-galactofuranose [10,11]. (**B**) The galactomannan exists under three forms: plasma membrane bound through a inositolphosphoceramide anchor [18], cell wall where the GM is cross linked to β-1,3-glucan [17] and secreted as a free polymer [9].

**Figure 2 jof-06-00283-f002:**
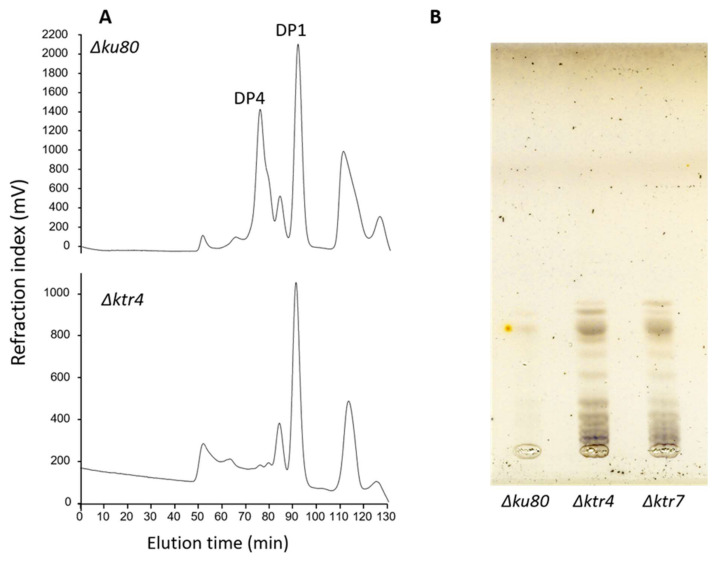
The structures of lipid anchored galactomannans are totally altered in *Δktr* mutants. (**A**) Analysis by acetolysis of the Lipo-galactomannan like produced by *Δktr4* mutant. Lipo-GM fraction was purified from total membrane of mycelium and purified as previously described [18,34]. Mannan structure was analysed by acetolysis degradation and gel filtration chromatography on TSK-40S column. The degradation of Lipo-GM from the parental strain (*Δku80*) leads to the release to two main products: DP1, corresponding to the galactofuranose that is sensitive to acetolysis and a DP4 corresponding to the tetramannoside repeat unit. Acetolysis cleaves preferentially α-1,6-mannosyl linkage. The degradation of the Lipo-GM produced by the *Δktr4* mutant did not release specifically a tetramannoside showing the absence of α-1,2-mannose linked residues. (DP: degree of polymerization). (**B**) TLC analysis of Glycosyl-Inositolphosphoceramide (GIPC) fraction produced by *Δktr* mutants. GIPC fractions was purified from from total membrane of mycelium and analysed by TLC (Thin layer chromatography), as previously described [36]. In comparaison to parental *Δku80* strain, both GM deficient mutants (*Δktr4* and *Δktr7*) produce larger GIPCs in higher amount.

**Figure 3 jof-06-00283-f003:**
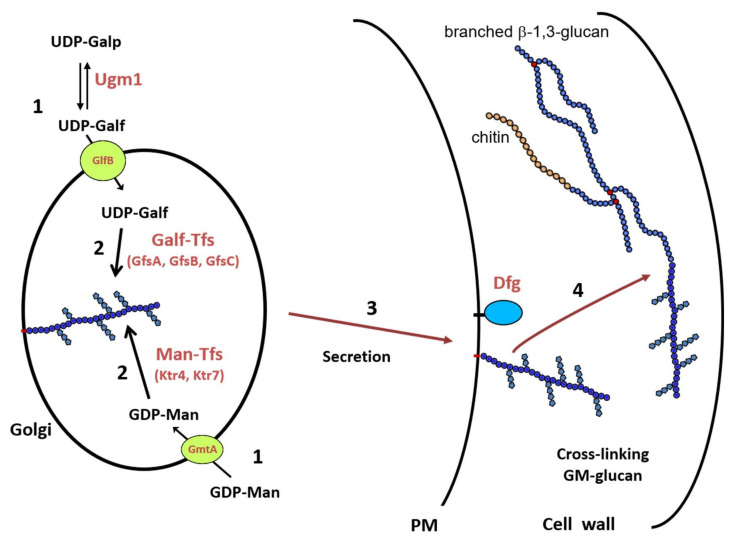
Biosynthetic scheme of the cell wall GM biosynthesis in *A. fumigatus*. (**1**) Synthesis of nucleotide-sugars in the cytosol and their transport into the lumen of the Golgi; (**2**) Polymerization of the GM by glycosyltransferases; (**3**) Intracellular trafficking of the lipo-GM to the plasma membrane; (**4**) Cross-linking of the GM onto cell wall β-1,3-glucan. Abbreviations: UDP-Galp: UDP-galactoyranose; UDP-Galf: UDP-galactofuranose; GDP-Man: GDP-mannose; Ugm1: UDP-galactose mutase; GmtA: GDP-Man transporter; GlfB: UDP-Galf transporter; GfsA,B,C: β-galactofuranosyltransferases; Ktr4,7: α-1,2-Mannosyltransferases.

**Figure 4 jof-06-00283-f004:**
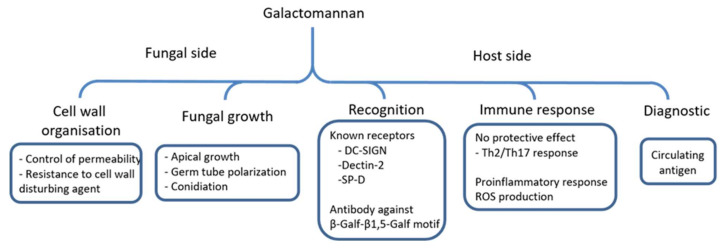
Functions of the GM produced by *A. fumigatus*. The scheme summarizes biological functions of the GM and its host interaction during infection.

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
