# Peer review of "Galactomannan Produced by Aspergillus fumigatus: An Update on the Structure, Biosynthesis and Biological Functions of an Emblematic Fungal Biomarker"

_jof, 2020, doi:10.3390/jof6040283_

Round 1
Reviewer 1 Report
Excellent and comprehensive review, reflecting the actual state of the affairs and open questions concerning the structure, biosynthesis and biological functions of GM.
Author Response
Dear reviewer,
Thank you for your kind comments on our review galactomannan produced by A. fumigatus. According to your remark, we read the manuscript carefully to improve the English language and to correct spelling mistakes.
Sincerely,
Thierry Fontaine

Reviewer 2 Report
The manuscript gives interesting insights about galactomannan. I have some comments.
- It would be helpful to add numbers to figure 3 showing visually the different steps of the pathway.
- Line 432 ‘Based on this incidence of aspergillosis infections’… What do the authors mean with ‘this incidence’?
- Improvement of the diagnostics tests. ‘Earlier detection of aspergillosis is required’. Will any of the knowledge presented in this manuscript allow the development of a more sensitive GM test? What do the authors mean with ‘defined structures produced by A. fumigatus during infection have never been investigated’?
- Line 372. There seems to lack one or more words ‘because of a (IL-4/Gata), and…’
- There are numerous typo’s in the text, e.g. line 42 ‘antidobies’, line 61 ‘varie’, line 65 Trychophyton,… and many more
- Explain what is meant with LFA-GM technology (line 43)
Author Response
Dear Reviewer,
Thank you for your kind comments on our review galactomannan produced by A. fumigatus. According to your remarks, we read the manuscript carefully to improve the English language and to correct spelling mistakes.
Steps of the biosynthesis are now numbered in the fig. 3
"Defined structures" has been replaced by Carbohydrate structures.
In perspectives, based on carbohydrate structure of the GM, we propose the improvement of the specificity of the detection. The increase of the sensitivity will be dependent on the affinity of antibodies recognizing mannan chain and galactofuranose side chain.
Line 432, incidence is now replaced by prevalence.
line 372 is now corrected
Sincerely
Thierry Fontaine
